# A Systematic Review on the Occurrence of *Babesia* spp. and *Anaplasma* spp. in Ticks and Wild Boar from Europe—A 15-Year Retrospective Study

**DOI:** 10.3390/pathogens14070612

**Published:** 2025-06-20

**Authors:** Ioan Cristian Dreghiciu, Diana Hoffman, Tiana Florea, Ion Oprescu, Simona Dumitru, Mirela Imre, Vlad Iorgoni, Anamaria Plesko, Sorin Morariu, Marius Stelian Ilie

**Affiliations:** 1Department of Parasitology and Parasitic Disease, Faculty of Veterinary Medicine, University of Life Sciences “King Mihai I” of Timisoara, 119, Calea Aradului, 300645 Timisoara, Romania; diana.hoffman@usvt.ro (D.H.); ionoprescu@usvt.ro (I.O.); mirela.imre@usvt.ro (M.I.); plesko.anamaria.fa@usvt.ro (A.P.); sorinmorariu@usvt.ro (S.M.); mariusilie@usvt.ro (M.S.I.); 2Veterinary and Food Safety Department 4, Surorile Martir Caceu, 300585 Timisoara, Romania; simonagiubega@gmail.com; 3Department of Infectious Diseases and Preventive Medicine, Faculty of Veterinary Medicine, University of Life Sciences “King Mihai I” of Timisoara, 119, Calea Aradului, 300645 Timisoara, Romania; vlad.iorgoni@usvt.ro

**Keywords:** *Babesia* spp., *Anaplasma* spp., wild boar, tick-borne pathogens, zoonotic diseases, Europe, molecular epidemiology

## Abstract

The wild boar (*Sus scrofa*) has experienced significant population growth as well as geographic expansion across Europe over the past 15 years, leading to increased concerns regarding its role in the transmission of zoonotic pathogens. Among these, *Babesia* spp. and *Anaplasma* spp. are of particular importance due to their impact on both wildlife and domestic animals. This study systematically reviews the prevalence and distribution of *Babesia* and *Anaplasma* spp. in wild boars and associated tick vectors across multiple European countries, synthesizing data from literature published between 2010 and 2024. A comprehensive search of Scopus, Google Scholar, and PubMed databases was conducted using predefined keywords related to babesiosis, anaplasmosis, wild boars, Europe, and tick-borne diseases. A total of 281 studies were initially retrieved, of which 19 met the inclusion criteria following relevance assessment. Data extraction focused on pathogen identification, diagnostic methods, sample type, host species, and prevalence rates. Molecular detection methods, primarily PCR and sequencing, were the most used diagnostic tools. Results indicate substantial regional variations in the prevalence of *Babesia* and *Anaplasma* spp. *A. phagocytophilum* was detected in wild boar populations across multiple countries, with the highest prevalence rates observed in Slovakia (28.2%) and Poland (20.34%). Conversely, lower prevalence rates were recorded in France (2%) and Portugal (3.1%). *Babesia* spp. showed higher prevalence rates in Italy (6.2%), while its detection in other regions such as Romania and Spain was minimal or absent. Notably, spleen and multi-organ samples (spleen/liver/kidney) exhibited higher positivity rates compared to blood samples, suggesting an organotropic localization of these pathogens. The findings underscore the role of wild boars as reservoirs for tick-borne pathogens and highlight their potential to contribute to the epidemiological cycle of these infections. The increasing distribution of wild boars, coupled with climate-driven shifts in tick populations, may further facilitate pathogen transmission. Future studies should focus on integrating molecular, serological, and ecological approaches to improve surveillance and risk assessment. Standardized methodologies across different regions will be essential in enhancing comparative epidemiological insights and informing targeted disease management strategies.

## 1. Introduction

Ticks serve as vectors for numerous parasitic and infectious diseases affecting both domestic and wild swine. In Europe, key tick-borne diseases impacting pigs include African swine fever, monocytic ehrlichiosis, babesiosis, anaplasmosis, and borreliosis [1].

The wild boar (*Sus scrofa*, L. 1758) is a highly adaptable and opportunistic species with a vast distribution range, spanning from Western Europe and the Mediterranean to Eastern Russia, Japan, and Southeast Asia. As one of the most widely dispersed terrestrial mammals, it thrives in diverse ecosystems, encompassing various vegetation types and climatic conditions (Figure 1) [2,3].

The European wild boar (*Sus scrofa*) has experienced a marked population and range expansion in recent decades due to multiple factors, including ecological adaptability, high reproductive capacity, rural depopulation, reintroduction efforts, absence of predators, changes in agriculture, reduced hunting pressure, feeding [5,6,7,8], economic impact and damage to agriculture and biodiversity [9,10], improved management strategies [5], landscape changes [11], and climate-related influences such as global warming and milder winters [12,13]. This expansion continues despite natural mortality caused by disease, extreme weather, and predation by wolves (*Canis lupus*) [6,14].

In order to provide a clearer overview of the wild boar population trends across Europe, below is a comparative figure that reflects the number of individuals hunted annually in several countries (Portugal [15], Spain [16], Sweden [17], and Poland [18]) between 2010 and 2024. This measure, often referred to as the “hunting bag”, represents the officially reported number of wild boars harvested during each hunting season and serves as a useful indicator of population dynamics and regional densities (Figure 2).

Babesiosis is a parasitic disease caused by protozoa of the *Babesia* genus, affecting both domestic and wild swine. The disease has a complex etiology involving multiple *Babesia* species known to infect pigs and wild boars. A review of the relevant scientific literature includes the following key species associated with porcine babesiosis:

*B. suis*—Recognized as the primary *Babesia* spp. responsible for babesiosis in domestic pigs, this parasite is transmitted via tick vectors such as *Ixodes ricinus* and *Dermacentor reticulatus*. Once inside the host, *B. suis* targets erythrocytes, leading to anemia and other clinical manifestations [19].

*Babesia* spp. MO1—A recently discovered *Babesia* spp., *Babesia* spp. MO1 has been detected in swine populations in China and Thailand. This species is suspected to be a significant pathogen in these regions, causing clinical signs such as anemia and jaundice, with transmission occurring through tick infestations [20].

*B. motasi*—Although identified in pigs across multiple Asian countries, *B. motasi* is considered relatively rare and less virulent compared to *B. suis* and *Babesia* spp. MO1 [19].

*B. bovis*—While capable of infecting pigs, *B. bovis* is regarded as a minor pathogen with lower virulence than other species within the genus *Babesia* [21].

Due to its often-nonspecific clinical presentation, porcine babesiosis can be overlooked, particularly when peripheral blood smears are not performed or when it is not included as a differential diagnosis in regions with a low prevalence of tick-borne infections. Symptoms such as jaundice, anemia, and hemoglobinuria are not exclusive to babesiosis, necessitating consideration of alternative diagnoses such as *Eperythrozoon suis* infection and acute leptospirosis [22].

Porcine anaplasmosis is an infectious disease caused by bacteria of the *Anaplasma* genus, with *A. phagocytophilum* being the predominant species involved. This pathogen is transmitted via hematophagous ticks that acquire the bacteria from infected animals, such as pigs, and subsequently transmit it to new hosts during blood meals. Clinical manifestations in pigs include fever, anorexia, lethargy, ocular inflammation, and weight loss, with severe infections potentially leading to mortality [23,24,25].

Numerous domestic (*Bos taurus*, *Sus domesticus*, *Canis familiaris*) and wild mammals (*Apodemus sylvaticus*, *Microtus agrestis*) have been identified as potential reservoirs of this bacteria in Europe [26].

*A. phagocytophilum* exhibits broad host adaptability and has been found in multiple genetic variants. Ongoing research focuses on the existence of subpopulations within the species and their implications for host specificity and pathogenicity [27].

This pathogen is a small, Gram-negative bacterium, approximately 0.4–1.3 µm in size. In mammalian hosts, *A. phagocytophilum* primarily targets neutrophils but is also capable of infecting both myeloid and non-myeloid cell types [28]. Within *Ixodidae* ticks, the bacterium persists in the salivary glands and midgut cells. Ultrastructural analyses have identified two morphological forms of *A. phagocytophilum*: the reticulate form and the dense-core (DC) form [29]. Proteomic studies indicate that these forms differ in protein expression by more than 20% when infecting human promyelocytic leukemia cells (HL-60). Experimental data suggest that the reticulate form represents the replicative but non-infectious stage of the bacterial life cycle, while the DC form, characterized by a condensed nucleoid structure, is more resistant to environmental changes and serves as the infectious stage for mammalian hosts [30].

While ticks can harbor numerous microorganisms, ranging from symbionts to pathogenic agents, detected through DNA analysis, their presence does not necessarily imply an infectious risk. This is due to several factors: the detection of pathogen DNA does not confirm the presence of viable, infectious organisms, and even when live pathogens are present, the tick may not be competent to transmit them [31].

Moreover, transmission of bacteria or parasites from infected ticks does not occur immediately; pathogens typically require time to mature and/or migrate to the salivary glands before transmission can take place, a process that may take up to 24 h [32].

Ticks often parasitize wild animals, which can act as reservoir hosts for various pathogens, thereby facilitating the maintenance and spread of infectious agents within natural ecosystems. Through blood feeding, ticks may acquire pathogens from these wildlife hosts, contributing to their circulation and potential spillover to domestic animals and humans [33,34].

This review aims to synthesize and critically evaluate the available scientific data from the past 15 years regarding the presence of *Babesia* spp. and *Anaplasma* spp. in wild boars and their associated tick vectors across Europe.

## 2. Materials and Methods

This systematic review was conducted in line with the recommendations of PRISMA 2020 (Preferred Reporting Items for Systematic Reviews and Meta-Analyses). The available literature on babesiosis and anaplasmosis was systematically reviewed using three independent databases—Scopus, Google Scholar, and PubMed—by applying a set of predefined search terms—*Babesia*, *Anaplasma*, wild boar, ticks, zoonoses, and Europe—using Boolean operators. No language restrictions applied. The research encompassed studies published between 2010 and 2024, focusing on the prevalence of zoonotic *Babesia* and *Anaplasma* spp. in animals and ticks.

Although literature prior to 2010 was considered during the initial screening, the review focused on studies published between 2010 and 2024 to better reflect current knowledge and research interest. A broader search using Google Scholar indicated that most relevant publications on *Babesia* spp. and *Anaplasma* spp. in wild boars and ticks have emerged within this time frame. Therefore, extending the period to include earlier decades would not have significantly impacted the outcomes of the review.

The summary table for animals categorized studies by publication year, country, disease detection method, host species, identified *Babesia* spp., total sample size, and prevalence. The articles were structured based on the country or location, demographics, medical history, and *Babesia* and *Anaplasma* spp. identified. Additionally, maps depicting reported cases were generated and georeferenced using MapChart version 6.8.1.

We included studies that

Were conducted in Europe;Analyzed wild boars and/or their associated ticks;Reported the prevalence of *Babesia* spp. and/or *Anaplasma* spp. infections.

Review articles, experimental studies, case reports, unpublished studies, and duplicates were excluded.

This review has not been previously registered in a public register such as PROSPERO, as no formal registration was foreseen at its inception.

## 3. Results and Discussion

The PRISMA 2020 flow diagram provides a summary of the research selection procedure [35]. From three databases, 282 records in all were obtained.

The search yielded 32 total results from Scopus (with 6 deemed relevant), 86 from PubMed (of which 10 were relevant), and 164 from Google Scholar (14 relevant) (Figure 3).

In total, 30 studies investigating babesiosis and anaplasmosis in animals or ticks were identified from Europe. However, ten studies were excluded to eliminate duplication. The remaining studies were systematically organized into a summary table.

Hence, we examined 20 papers.

Over the past 15 years, most studies conducted in Europe to detect *Babesia* spp. and *Anaplasma* spp. in wild boars and their attached ticks have originated from Italy.

The following map (Figure 4) illustrates the distribution of *Anaplasma* and *Babesia* spp. in different European countries, categorized as follows:Green (Absent): Countries where no *Anaplasma* spp. or *Babesia* spp. have been detected (e.g., Spain).Yellow (*Anaplasma* spp.): Countries where *Anaplasma* spp. has been detected but not *Babesia* spp. (e.g., Poland, Germany, Romania).Patterned Red (*Anaplasma* spp. + *Babesia* spp.): Countries where both *Anaplasma* spp. and *Babesia* spp. have been detected (e.g., France, Italy, Hungary).

The map provides a visual representation of the geographical prevalence of these tick-borne pathogens, highlighting regions with a higher risk of infection.

**Figure 4 pathogens-14-00612-f004:**
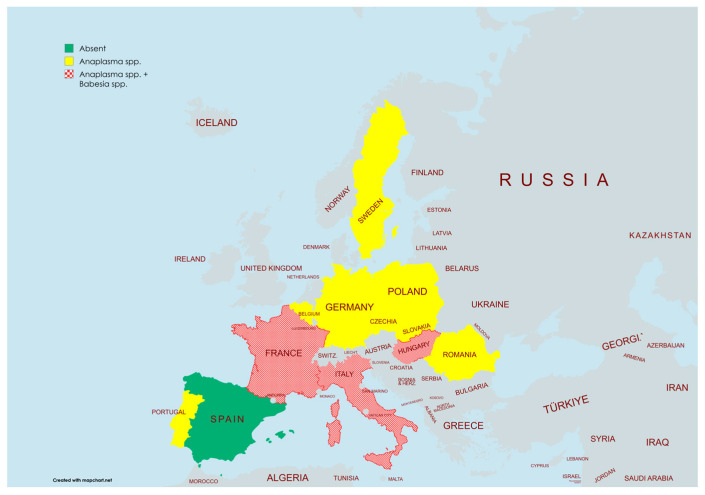
Geographical distribution of *Anaplasma* and *Babesia* spp. in Europe [36].

The following tables (Table 1 and Table 2) provide a comprehensive overview of studies investigating *Anaplasma* and *Babesia* spp. in wild boars and ticks across various European countries. They include details such as the year of study, country, diagnostic methods, sample type, identified species, total number of analyzed samples, number of positive cases, and prevalence rates.

As PCR was consistently used as the diagnostic method across all studies, it was not specifically indicated in the table; the letter ‘S’ was added only in cases where PCR was followed by sequencing.

The reported prevalence of *Anaplasma* spp. in wild boar or in ticks collected from these animals has varied greatly across European countries over the past 15 years, reflecting differences in ecological conditions, host density, vector distribution, sampling strategies, and diagnostic methods employed. Some studies have reported relatively low prevalence rates, while others have documented significantly higher levels of infection, suggesting that regional factors, such as climate, habitat type, and wildlife management practices, may strongly influence the circulation and detection of these pathogens.

For formatting purposes, *Anaplasma phagocytophilum* was abbreviated as *A. phag.* in the table, as only a few species were identified for this pathogen.

Prevalence for *Anaplasma* spp. rates have ranged from as low as 0% in Spain [43] to 28.20% in Slovakia [57], and for *Babesia* spp., the prevalence ranges from 0% in Spain [43] to 1.50% in Hungary [38].

*A. phagocytophilum* was found in 5 of the 513 spleen samples that were analyzed in Belgium. These infections were determined by sequencing analysis to be members of the Dama 35 strain (GenBank accession number GQ450276.1) and the CE18 strain (GenBank accession number GQ450278.1). Compared to the previously reported prevalence of 85.6% in roe deer from the same location, the total prevalence in wild boars was 0.97% (95% CI: 0.12–1.82) [46].

Similar to these findings, *A. phagocytophilum* has been reported at low prevalence levels in other countries, such as the Czech Republic, where 5.10% (28 out of 550 samples) tested positive [47], and Romania (2.96%, 3.44%, 4.48%) [42,55,56].

Several studies from mainland France and Corsica have reported the presence of *Anaplasma* spp. and *Babesia* spp. in wild boars and ticks collected from them. In central France, Dugat et al. (2017) detected *A. phagocytophilum* in 44.8% of wild boar spleen samples collected between 2009 and 2015, the highest prevalence reported in the country [48]. Ticks collected from wild boars in the same region, including *Rh. bursa* and *D. marginatus*, were also found to carry *Anaplasma* spp., suggesting their role as potential vectors [37,38].

In Corsica, Grech-Angelini et al. (2019) analyzed ticks from wild and domestic animals, identifying *A. phagocytophilum* DNA in 2% of tick pools collected from wild boars [49]. The identified vector species included *Rh. bursa*, *D. marginatus*, *I. ricinus*, *Hyalomma marginatum*, and *R. sanguineus*.

More recently, Defaye et al. (2021) screened 158 wild boars from Corsica and reported three positive samples for *Babesia* spp., although sequencing results were inconclusive, preventing precise species identification [37]. Among the 113 ticks collected from these wild boars, 79.6% were positive for at least one vector-borne pathogen, and one *D. marginatus* tick tested positive for *Anaplasma* spp., though sequencing could not confirm the species [37].

In France (Corsica), the presence of *Anaplasma* spp. was minimal, with a prevalence rate of 0.88% (10/113 samples) [37]. Similarly, *A. phagocytophilum* in Corsican samples exhibited a low prevalence of 2% (4/177 samples) [49].

These findings suggest that both *Anaplasma* and *Babesia* spp. circulate in wild boars and their associated ticks in France, with marked regional differences in prevalence. The results also highlight the need for further studies to clarify the role of wild boars in the transmission of these pathogens and the involvement of various tick vectors across different French ecosystems.

Reported infection rates with *A. phagocytophilum* in wild boars from other regions have ranged between 0% and 28.2%. Higher prevalence levels have been observed in southern Germany (where 12.50% of sampled wild boars tested positive for *A. phagocytophilum*) [50], Poland (20.34%) [52], and Slovakia (28.2%) [57]. In contrast, studies conducted in Austria and northern Spain did not detect any wild boars infected with *A. phagocytophilum* [43,58,59].

*A. phagocytophilum* was found to be moderately prevalent in Hungary, with a frequency of 5.7% (19/333). However, just 5 out of 333 samples tested positive (1.5%) for *B. canis* and *B. crassa*, indicating a much lower frequency in Hungary [38].

Between 2010 and 2022, several studies reported the presence of *Anaplasma* and *Babesia* spp. in wild boars from different regions of Italy. In central Italy, Ebani et al. (2017) analyzed 100 spleen samples collected during hunting seasons between 2013 and 2015, detecting *A. phagocytophilum* in one sample (1%)—the first report of this pathogen in wild boars from Italy [51].

In northern Italy, Zanet et al. (2014) examined over 1000 wild ungulates, including 257 wild boars, and detected *B. bigemina* in 4.67% of wild boar samples using nested PCR and sequencing [39].

In southern Italy, Sgroi et al. (2023) analyzed 243 spleen samples collected between 2016 and 2022 and identified *B. vulpes* (5.3%) and *Babesia capreoli* (0.9%), with a total prevalence of 6.2% [41]. Although *B. vulpes* was not detected in blood samples, the presence of its suspected vectors (*Ixodes hexagonus*, *Ixodes canisuga*) [60,61], and that of *I. ricinus* for *B. capreoli* [62,63], suggests potential vector–host interaction in the area. These ticks are common on wildlife such as foxes, which share habitats with wild boars and may play a role in maintaining the transmission cycle [41].

In contrast, Zobba et al. (2014) found no evidence of *Babesia* spp. in 52 blood samples collected from wild boars in Sardinia [40]. The absence of positive cases highlights regional differences in prevalence across mainland Italy and the islands.

In Portugal, 141 free-ranging ungulates, comprising 73 red deer (*Cervus elaphus*), 65 wild boars (*Sus scrofa*), and 3 fallow deer (*Dama dama*), sampled from both sexes, were studied throughout the shooting seasons of December 2013 to March 2015. Five districts, Castello Branco (n = 31), Portalegre (n = 16), Lisboa (n = 19), Évora (n = 15), and Beja (n = 60), were used to collect the animals.

Two wild boars (3.1%; 95% CI: 0.4–10.7%) that were sampled had co-infections with *Theileria capreoli* and *A. platys*. Furthermore, red deer from Castelo Branco had a considerably higher incidence of *Anaplasma* infection than those from Beja (*p* = 0.019). Interestingly, the Beja district was the source of all wild boar blood samples that tested positive for *T. capreoli* or *A. platys* [54].

The locations that were sampled showed varying rates of pathogens discovered in ticks. With 93 out of 484 samples testing positive for *A. phagocytophilum*, Poland (East) had the greatest number of positive cases, with a 19.2% prevalence rate [52]. Germany (South), on the other hand, had the greatest relative prevalence (25%), even though there were only 16 samples, 4 of which tested positive [50].

Several studies have investigated the presence of *A. phagocytophilum* and *Babesia* spp. in wild boars from different regions of Romania. Kiss et al. (2014) conducted a large-scale survey between 2007 and 2012 on 870 organ samples (spleen, liver, kidney) collected from wild boars across 16 counties. Using nested PCR targeting the 16S rRNA gene, *A. phagocytophilum* was detected in 4.48% of samples, with most positive cases concentrated in central Transylvania [55].

Matei et al. (2023) analyzed 203 blood samples collected during two hunting seasons (2019–2020 and 2020–2021) in Sălaj County. They identified *A. phagocytophilum* in 3.0% of samples using conventional and nested PCR targeting both the 16S rRNA and groEL genes. No samples tested positive for *Babesia* spp. [42].

In another recent study, Dreghiciu et al. (2023) examined twenty-nine blood samples from wild boars in Hunedoara and Timiș counties, detecting *A. phagocytophilum* in one case (3.44%) using PCR targeting the epank1 gene [56].

Although prevalence rates remain relatively low, these findings confirm the circulation of *A. phagocytophilum* in wild boars from multiple regions of Romania. The absence of *Babesia* spp. in tested samples suggests limited distribution or lower host susceptibility, but further studies are needed to clarify this aspect.

In a Swedish study focusing on wild boars, spleen samples from 34 individuals were analyzed to detect the presence of tick-borne pathogens. The results revealed that *A. phagocytophilum* was present in 24 animals, indicating a notable infection rate of 70.6%. On the other hand, *Babesia* spp. was not detected in any of the samples tested [44]. These findings point toward a significant exposure of wild boars to *A. phagocytophilum* in this region, while suggesting a minimal or absent role in the transmission of *Babesia* under the conditions of this investigation.

### General Discussion

This review provides an updated overview of the presence of *Babesia* spp. and *Anaplasma* spp. in wild boars and their associated ticks across Europe, based on studies published between 2010 and 2024. We acknowledge that differences in reported prevalence may be due to both real epidemiological variation and differences in study design or sampling. As these factors are often difficult to separate, this limitation has been noted. Even so, the collected data offers useful insights into the distribution of these pathogens and the potential role of wild boars and ticks in their circulation.

The hypothesis that wild boars can contract spontaneous infections but can also efficiently manage the pathogen is supported by the low incidence of *A. phagocytophilum* in these animals. Innate immune response activation and cytoskeletal reorganization are two potential mechanisms for this control, which might promote improved phagocytosis and autophagy and slow the spread of infection [64]. These findings demonstrate the geographical differences in pathogen dispersion, with France (Corsica) showing the lowest pathogen detection frequencies and Poland and Germany showing the greatest prevalence rates. Tick vector distribution, host reservoir dynamics, or environmental conditions may have an impact on the observed discrepancies, requiring more research (Figure 5).

Significant geographical diversity in infection rates was shown by the range of *A. phagocytophilum* prevalence in ticks, which varied from a low of 0.88% in Corsica (France) [37] to a maximum of 25% in southern Germany [50].

Spleen, liver, and kidney samples from wild boars are among the most informative for disease diagnosis, owing to both their higher sample counts and elevated positivity rates. In contrast, although numerous blood samples were collected, their relatively low prevalence rates suggest limited diagnostic value. As illustrated in Figure 6, spleen samples and combined multi-organ samples (spleen, liver, and kidney) appear to be the most reliable for detecting infections.

The examination of pathogen presence in several wild boar sample types showed notable differences in frequency, positive rate, and sample size. With the biggest sample size (n = 870) and the highest number of positive cases (n = 39), the wild boar spleen/liver/kidney group demonstrated a significant presence of pathogens in multi-organ samples.

The identification of pathogens in wild boars differed depending on the pathogen and tissue type. The efficacy of multi-tissue sampling was demonstrated by the highest prevalence of *A. phagocytophilum* in spleen/blood (28.20%) [57] and spleen/liver (20.34%) [52]. Other pathogens such as *Babesia* spp. showed very low to no detection across all sample types [37,38,39,40]. Overall, compared to blood alone, spleen and mixed tissue samples demonstrated greater reliability for pathogen surveillance.

This result implies that compared to spleen or multi-organ samples, blood samples can be less effective in identifying specific pathogen samples.

Until 2010, there was a clear lack of research focused on *Babesia* infections in wild boars or in ticks collected from them. The topic was rarely addressed, and detailed molecular studies were mostly missing. This made it difficult to understand the role that wild boars and their ectoparasites might have in the transmission cycle of *Babesia* spp. Overall, the limited data available from that period highlight how little was known about the involvement of wildlife in the ecology of these pathogens.

Despite its broad geographic range, considerable gaps remain in our understanding of *A. phagocytophilum* ecology, epidemiology, and vector–host dynamics. Limited data on the biological traits of its vectors, the diversity and pathogenic potential of circulating strains, and the infection sources continue to prevent the development of targeted strategies for disease control and prevention in both animals and humans [65]. The lack of evidence for transovarial transmission in the main tick vectors linked to human and animal infections highlights the critical role of vertebrate reservoirs in sustaining the pathogen in natural ecosystems [66,67]. Therefore, improving our understanding of strain variability, spatial distribution, and host–vector–pathogen interactions is essential for producing accurate risk assessments and informing veterinary and public health interventions.

Although traditionally considered a single bacterial species, *A. phagocytophilum* has shown considerable genetic diversity and a broad host range, as revealed by 16S rRNA gene analyses [68]. The detection of specific genetic variants in ticks appears to correlate with habitat characteristics and the presence of certain reservoir hosts, suggesting some degree of host-linked strain distribution. When more refined molecular approaches are applied, patterns of host specificity become increasingly evident, pointing to a complex network of host–pathogen interactions. These findings underscore the importance of considering the genetic variability of *A. phagocytophilum* when evaluating its transmission pathways, particularly the potential involvement of wild ruminants in maintaining and spreading strains that may also affect domestic animals and humans [69].

Studies conducted in Central Europe until 2010 have reported variable prevalence rates of *A. phagocytophilum* in wild boar tissues. In Poland, Michalik et al. (2012) identified the pathogen in 12% of 325 spleen samples using nested PCR [70]. In Slovakia, Smetanová et al. (2006) reported a prevalence of 5.5% based on 18 samples [71], while in the Czech Republic, Hulínská et al. (2004) found a detection rate of 4.4% in 69 wild boars [72]. A study from Slovenia conducted by Galindo et al., 2013, also found a relatively low prevalence of 2.7% in 113 wild boar samples [64]. These findings indicate a relatively low to moderate presence of *A. phagocytophilum* in wild boars across these regions.

In contrast, more recent investigations included in our review show notably higher rates, such as 20.34% in Poland [52], 28.2% in Slovakia [57], 44,8% in France [48], and 70.60% in Sweden [44]. Meanwhile, several countries such as Portugal (3.1%) [53] and Romania (2.96–4.48%) [42,55,56] continue to report consistently low levels of infection. These findings suggest both ecological and methodological sources of variation and emphasize the need for standardized diagnostic approaches to enable reliable comparisons between regions.

Prevalence studies of *A. phagocytophilum* in questing ticks across Europe show a wide range of values, as reported in multiple countries using PCR-based methods. While most prevalence estimates fall between 1% and 10%, a few countries report markedly higher or lower rates. For example, Denmark recorded one of the highest prevalences at 23.6% [73], and Bulgaria reported 33.9% in adult ticks, contrasting with just 2.2% in nymphs [74]. In Italy, values ranged from 1.5% [75] to 24.4% [76], reflecting regional and temporal variability. Conversely, countries such as Hungary (0.4%) [77], the Netherlands (0.6%) [78], Luxembourg (1.9%) [79], and Switzerland (1.2–2%) [80,81,82] consistently showed low infection rates. Germany presented predominantly low to moderate values, between 1.0% and 5.4%, though isolated studies reported higher figures up to 17.4%. Other countries, including Poland, France, and the UK, exhibited intermediate prevalence rates typically in the 2–10% range, although with noticeable intra-national variation [69].

These differences likely stem from variations in geographic region, tick developmental stage, detection method (PCR vs. qPCR), and host or reservoir presence. Collectively, the data highlights the uneven but widespread distribution of *A. phagocytophilum* in European tick populations.

When comparing the prevalence of *Anaplasma phagocytophilum* in questing ticks, as compiled by Stuen et al. (2013) [69], with the data from our review referring to ticks collected from wild boars, similar trends in variability are observed. In both reviews, prevalence values range from very low (e.g., 0.88–2% in France (Corsica) [37,49] and 5.70% in Hungary [38]) to relatively high prevalence, such as 19.2% in Poland and 25% [53] in Germany [50], which is consistent across both questing ticks and host-attached ticks. These parallels suggest comparable exposure prevalence in certain geographic areas. Differences between the two groups are also notable and may be influenced by the stage of the tick, the timing of sample collection, or differences in host availability and reservoir competence.

Overall, both datasets confirm the widespread yet not equal presence of *A. phagocytophilum* in European tick populations, whether questing or associated with wild boars.

It is important to recognize the limitations of this review. A formal risk of bias assessment was not conducted, and a meta-analysis could not be performed due to the heterogeneity of study designs and diagnostic techniques. Despite these limitations, the synthesis identifies important regional and epidemiological trends and offers a thorough summary of the incidence of *Babesia* spp. and *Anaplasma* spp. in wild boars and ticks throughout Europe.

We acknowledge that some of the variation in reported prevalence may be due to differences in study design, sampling methods, and molecular techniques. It is also possible that these differences reflect true variations in pathogen circulation by ecological, geographical, and host-specific factors.

## 4. Conclusions

The review demonstrates that wild boars across Europe serve as significant hosts for a diverse range of vector-borne pathogens, including *Anaplasma* and *Babesia* spp. Vector ticks, including *D. marginatus* and *Rh. bursa*, were frequently infected, suggesting that they may play a role in the transmission dynamics, which is not usually observed in *I. ricinus*.

The consistent detection of zoonotic agents, especially *A. phagocytophilum* and *Babesia* spp., across various studies highlights their potential role in maintaining and possibly amplifying these pathogens in natural environments. The reviewed literature also reveals substantial methodological heterogeneity, with differences in sample types (blood, spleen, ticks), molecular protocols, and reporting standards, which limits direct comparison between studies. To enhance epidemiological insight and inform public and veterinary health strategies, future research should prioritize harmonized sampling procedures and standardized molecular diagnostics.

These findings also underscore the need for standardized methodologies to improve data comparability across Europe.

## Figures and Tables

**Figure 1 pathogens-14-00612-f001:**
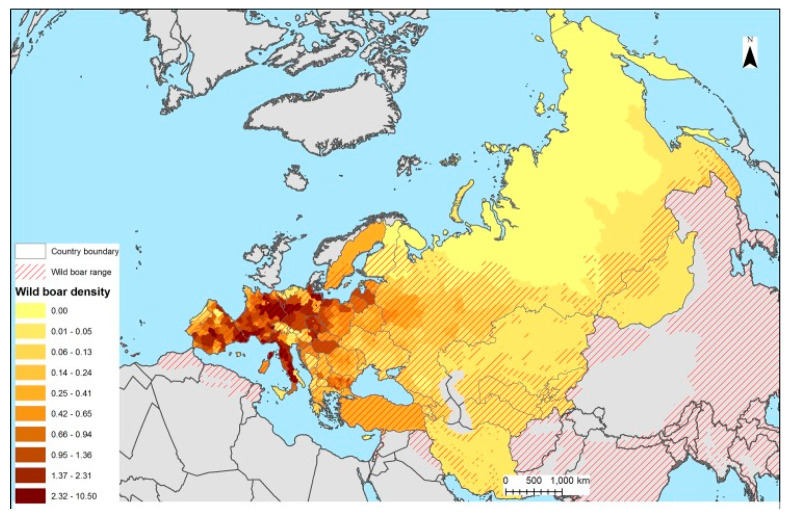
Wild boar density in Europe and Asia by Pittiglio et al., 2018 [4].

**Figure 2 pathogens-14-00612-f002:**
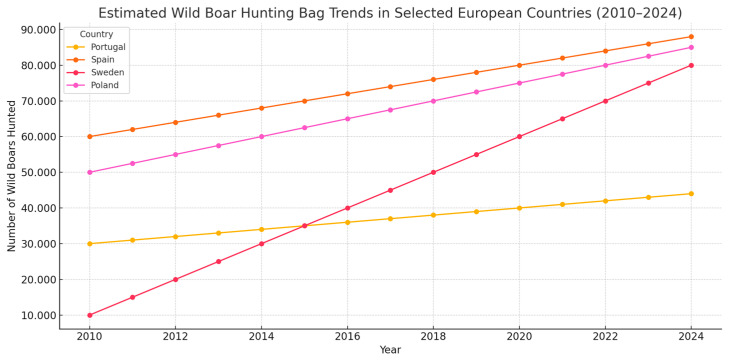
Estimated wild boar hunting bag in selected European countries (2010–2024). Portugal [15], Spain [16], Sweden [17], Poland [18].

**Figure 3 pathogens-14-00612-f003:**
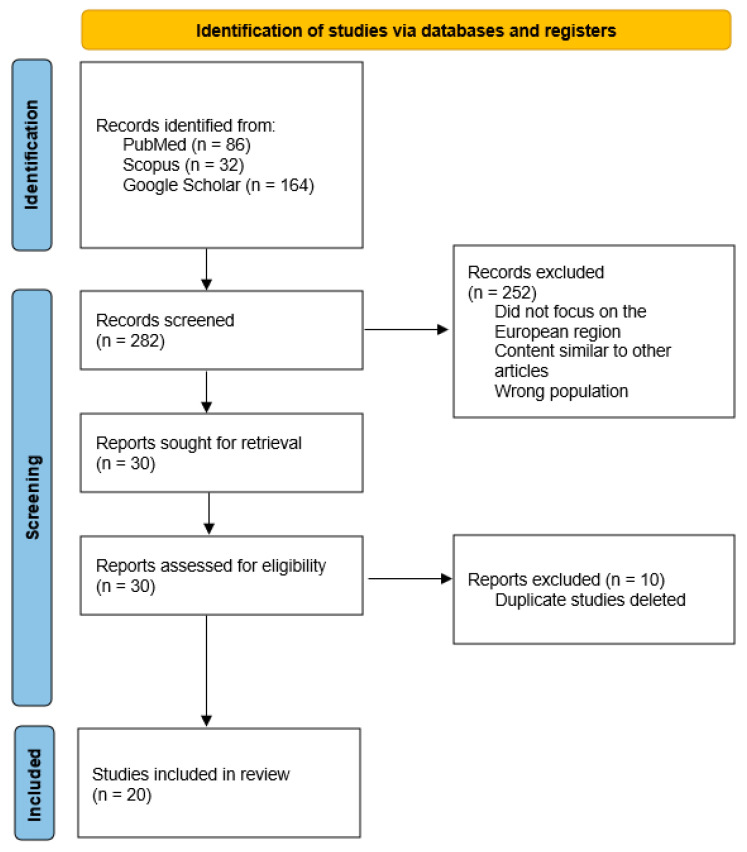
Identification and screening of studies on tick-borne pathogens in European wild boars.

**Figure 5 pathogens-14-00612-f005:**
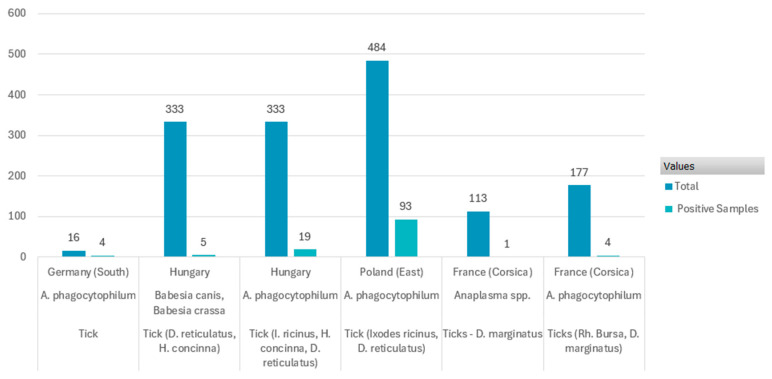
Prevalence of pathogens in ticks from wild boar.

**Figure 6 pathogens-14-00612-f006:**
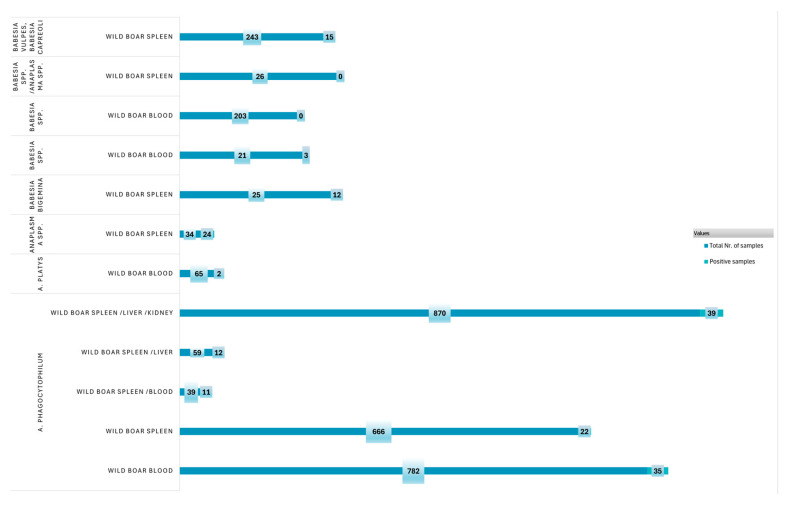
Prevalence of pathogens in wild boar samples.

**Table 1 pathogens-14-00612-t001:** Evidence of *Babesia* in wild boars and ticks across European countries.

Years of Study	Country	Diag. Method	Host/Sample	Species	No. of Samples	Positive	Prevalence	Ref.
2018–2020	France (Corsica)		Blood	*Babesia* spp.	158	3	** *2%* **	[37]
2020	Hungary		Ticks (*D. reticulatus*, *H. concinna*)	*B. canis*, *B. crassa*	333	5	** *1.50%* **	[38]
2008–2012	Italy (North)		Spleen	*B. bigemina*	257	12	** *4.67%* **	[39]
2010–2013	Italy (Sardinia)		Blood	*Babesia* spp.	52	0	** *0%* **	[40]
2016–2022	Italy (South)	S.	Spleen	*B. vulpes*, *B. capreoli*	243	15	** *6.20%* **	[41]
2019–2021	Romania	S.	Blood	*Babesia* spp.	203	0	** *0%* **	[42]
2009–2015	Spain		Spleen	*Babesia* spp.	269	0	** *0%* **	[43]
2018–2019	Sweden		Spleen	*Babesia* spp.	34	0	** *0%* **	[44]
2013–2014	Sweden		Ticks (*I. ricinus*)	*B. microti*,	519	17	** *3.6%* **	[45]
				*B. divergens*,		1	** *0.2%* **	
				*B. venatorum*		5	** *1%* **	

*Legend*: *PCR was consistently used as the diagnostic method across all studies*; *therefore*, *it is not specifically indicated in the table*; *S* = *Sequencing*.

**Table 2 pathogens-14-00612-t002:** Evidence of *Anaplasma* spp. in wild boars and ticks across European countries.

Years of Study	Country	Diag. Method	Host/Sample	Species	No. of Samples	Positive	Prevalence	Ref.
2011	Belgium		Spleen	*A. phag.*	513	5	** *0.97%* **	[46]
2018–2019	Czech Rep.	S.	Blood	*A. phag.*	550	28	** *5.10%* **	[47]
2009–2015	France (Central)		Spleen	*A. phag.*	29	13	** *44.80%* **	[48]
2014–2015	France (Corsica)		Tick (*Rhipicephalus bursa*, *D. marginatus*)	*A. phag.*	177	4	** *2%* **	[49]
2018–2020	France (Corsica)		Tick (*D. marginatus*)	*Anaplasma* spp.	113	1	** *0.88%* **	[37]
2010–2013	Germany (South)	S.	Spleen	*A. phag.*	24	3	** *12.50%* **	[50]
2010–2013	Germany (South)	S.	Tick	*A. phag.*	16	4	** *25%* **	[50]
2020	Hungary		Tick (*I. ricinus*, *H. concinna*, *D. reticulatus*)	*A. phag.*	333	19	** *5.70%* **	[38]
2013–2015	Italy (Central)		Spleen	*A. phag.*	100	1	** *1%* **	[51]
2017–2019	Poland	S.	Spleen/liver	*A. phag.*	59	12	** *20.34%* **	[52]
2018–2020	Poland (East)		Tick (*I. ricinus*, *D. reticulatus*)	*A. phag.*	484	93	** *19.20%* **	[53]
2013–2015	Portugal	S.	Blood	*A. platys*	65	2	** *3.10%* **	[54]
2019–2020	Romania	S.	Blood	*A. phag.*	203	6	** *2.96%* **	[42]
2007–2012	Romania (Central)		Spleen/liver/kidney	*A. phag.*	870	39	** *4.48%* **	[55]
2023	Romania (West)		Blood	*A. phag.*	29	1	** *3.44%* **	[56]
2011–2014	Slovakia		Spleen/blood	*A. phag.*	39	11	** *28.20%* **	[57]
2009–2015	Spain		Spleen	*Anaplasma* spp.	269	0	** *0%* **	[43]
2018–2019	Sweden		Spleen	*Anaplasma* spp.	34	24	** *70.60%* **	[44]

*Legend*: *A. phag.* = *A. phagocytophilum*; *A. phag. was used as an abbreviation for Anaplasma phagocytophilum in the table*; *PCR was consistently used as the diagnostic method across all studies*; *therefore*, *it is not specifically indicated in the table*; *S* = *Sequencing*.

## Data Availability

Data are contained within the article.

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
