# Peer review of "A Systematic Review on the Occurrence of Babesia spp. and Anaplasma spp. in Ticks and Wild Boar from Europe—A 15-Year Retrospective Study"

_pathogens, 2025, doi:10.3390/pathogens14070612_

Round 1
Reviewer 1 Report
Comments and Suggestions for Authors
This review underscores the importance of continuous surveillance of Babesia and Anaplasma species in wild boars and relevant ticks to monitor emerging trends and potential zoonotic threats. However, the manuscript appears insufficient to be published in the currently status. Minor revisions are required.
- The scientific name of both tick and pathogens should be spelled in the standard manner. For example, the first appearance should be written in full spelling of Genus + species style, and the followings can be written in abbreviation of genus name + species style. And all scientific name should be written in italic style.
- In the part of introduction, the reasons for the increased wild boar population were written with too many words, more concise expression or rephrase would be appreciated.
- The contents about tick species parasite on wild boar should be addressed since the topic of the manuscript focused on the tick borne Babesia or Anaplasma pathogens. More detailed information provided would be acceptable.
- The comerical trade mark such as TM or others are suggested to omit in the manuscript to avoid the potential conflict of interests.
Reviewer 2 Report
Comments and Suggestions for Authors
Comments for” A review on the occurrence of Babesia spp. and Anaplasma spp. in ticks and wild boar from Europe -15 years retrospective
General comments.
The paper describes a literature study aiming at assessing the geographical differences in Babesia and Anaplasma prevalence in wild boars in Europe. A literature search was conducted, and relevant papers were identified. The methods and outcome of the studies were described and condensed into maps and tables in a mixed results and discussion section.
Introduction: The introduction is OK although it would be prudent to show a map of Wild Boar populations in European countries (or the hunting bag), such that the readers have a clear idea of what is going to follow. Materials and methods: This section is OK, but they need to explain why the period 2010-2025 was chosen. Results and Discussion: The result and discussion section sets off by describing studies in Italy, France, and Romania in great detail, which is followed by a far less detailed section on other countries. Conclusion: OK
Overall assessment.
The idea and scope of the paper are ok, but the authors seem to have a vague sense of direction – and thus the purpose is unclear. This is also reflected in highly uneven reporting from the different countries and a general discussion that lacks momentum.
Specific comments.
Line 66 – or there about. You are providing a lot of information on the wild boar population in the text but it is difficult to take in. I recommend that you insert a map showing the distribution/densities of wild boars in Europe and a graph showing population development e.g. change in the size of the hunting bag in various countries.
Line 100: you write that the white-footed mouse is the key host, which is quite unfortunate since the paper is set in a European setting. White-footed mice only live in North America.
Line 117-119: a paragraph that seems unfitting as a closure of an introduction. The end of the introduction should be reserved for a statement of the objective/hypothesis of the study.
Line 127: pls. Explain why you did not extract papers from e.g. 2000-2024, and inform the readers that it would not make much difference if you did. You can use the Google Scholar counts as a reference – i.e. show the number for 1990 to 2024, 2000-2024, etc.
Line: 151: Fig 1 – Please add the causes for exclusion to the relevant box, and write something more informative in the legend than – “Flow chart”.
Line 181: Table 1. The table seems to list the recovered information and faithfully includes the redundant comment that the continent is “Europe”. The column can be deleted.
It also includes information recovered from the analysis of ticks, even though this seems out of place in the given study. It is not given that the babesia recovered in the tick originates from the given host as Babesia can be transmitted transovarially. You will either have to remove these or add in papers on ticks from other countries e.g., https://www.sciencedirect.com/science/article/abs/pii/S1877959X15001466
The given records do unexpectedly not include Sweden although the relevant studies have been performed: https://parasitesandvectors.biomedcentral.com/articles/10.1186/s13071-021-04860-w
Please consider splitting the table in two: one for babesia and one for anaplasma.
Line 192, 252, 302. – re. Italy, France, Romania – The description is far too detailed and includes a large amount of information that is irrelevant to the context. The level of detail should be reduced to that applied in section 3.4 – other countries. Pls. organize the text such that it follows the (alphabetical) logic used in Table 1.
Line 342 – should begin a new section as it deals with the totality of the observations. The section needs to embrace the wider aspects of the study. Importantly the authors need to determine whether the observed differences are linked to differences in methodology or whether they reflect true differences in transmission. If they are unable to conclude whether it is one or the other – then it must be clearly stated. In support of this, I recommend that they refer to other studies such as Stuen, S., Granquist, E. G., & Silaghi, C. (2013). Anaplasma phagocytophilum—a widespread multi-host pathogen with highly adaptive strategies. Frontiers in cellular and infection microbiology, 3, 31., such that they can determine whether the findings are consistent or inconsistent with previous studies.
Round 2
Reviewer 2 Report
Comments and Suggestions for Authors
Comment to revised version of: A review on the occurrence of Babesia spp. and Anaplasma spp. in ticks and wild boar from Europe -15 years retrospective
General comments
The authors made substantial changes to the manuscript, which in many cases adhered to the comments.
There do however seem to be some discrepancies between their rebuttal and the actual changes made. Responses 1, 3, 4, 5, 6, 8, 9, 10 are OK. Response 2 is So, so – since the North American rodent is still mentioned in the text. Oddly, no genus names are given for the European rodents (Apodemus, Microtus, etc.). Response 7. The authors are avoiding the issue and Response 11. There is no new section as indicated in the rebuttal, and the end of the paper still lags structure and momentum.
The lack of consistency, typographical errors, and lack of contextual “clean up” in the revision suggest that it has been done too hastily. So pls. let the author take time in the next round.
Importantly, the authors must ensure that the results are presented unambiguously, and that there is no uncertainty what “the prevalence” refers to. Moreover, the structural problems remain, and the clarity of the message needs to be improved.
Specific comments.
Line 46: I don’t think you can talk about “crucial” vectors. Pathogens are typically either transmitted by these or not, so it is pointless to say that they are “crucial”.
Line 49-61: This paragraph seems out of place – it is a bit early to mention the difficulties in the interpretation – before you framed the overall questions. – move to after line 130 or the general discussion.
Line 66: Figure not figure.
Line 82: Figure not figure.
Line 84: list the references in the legend.
Line 115: delete “however”.
Line 143: The sentence “The search yielded 32 total results from Scopus (with 143 deemed relevant), 86 from PubMed (of which 10 were relevant), and 164 from Google 144 Scholar (14 relevant) (figure 1). – belongs in the results around line 170. Figure not figure.
Line 152: In total, 30 studies investigating babesiosis and anaplasmosis in animals or ticks were identified from Europe. However, ten studies were excluded to eliminate duplication. The remaining studies were systematically organized into a summary table. – belongs partly in the results around line 170.
Line 168: move the heading Results and Discussion to the top of page 5.
Line 193: write “Tables 1 and 2”
Line 198: the sentence “Due to the heterogeneity of the included studies in terms of diagnostic methods and sample types, a meta-analysis was not performed. Results were summarized narratively and table summarized” belongs in the M&M section. – possibly just delete.
Line 202: Widen Table 1 and check that the word in the heading is split appropriately.
Line 204: Widen Table 2 and check that the word in the heading is split appropriately.
For Tables 1 and 2. Delete the word Wild boar from the column (host) – just mention the tissues blood, spleen, and ticks. Delete the word PCR and just make a sign e.g. S, when the PCR was followed by sequencing (which will make the column narrower). Possibly, Abbreviate A. phagocytophilum such that it does not generate multiline-rows, and explain this in the legend.
Please get rid of the strange coloring of the tables and move some text in between the two tables.
Line 206- 215: I do not see the point of these paragraphs – delete?
Line 217. There is a “].” lost in the line. Delete.
Lines 224-226. Does not convey any information. Delete.
Line 264- 268: is a general commentary and belongs to the overall discussion.
Line 300. You write “The locations that were sampled showed varying levels of the disease discovered in ticks. With 93 out of 484 samples testing positive.” But I have no idea what they were positive for – Anaplasma or Babesia? – the same applies to a few other paragraphs.
Also..the samples do not have “disease”, they inform the presence of pathogens!
Line 322 and 323 – are void of content – delete.
Line 330. Make the heading “General discussion”
Line 432: Figure not figure.
Figure 4 The legend is inadequate e.g. for “Sum of prevalence ?? – what do the numbers mean??
Line 378: The conclusion only contains some very general commentary suggesting that nothing was learned in the process of the review. Please read your discussion and just state what the essence is – and reduce the amount of general commentary.
If you can write “ we acknowledge that some of the variation in prevalence may be due to differences in methodology. …., it's also possible that these differences reflect real variations in pathogen circulation….” In your rebuttal, then surely you can conclude the same in the paper.
Round 3
Reviewer 2 Report
Comments and Suggestions for Authors
Comment to second revised version of: A review on the occurrence of Babesia spp. and Anaplasma spp. in ticks and wild boar from Europe -15 years retrospective
General comments
The authors made substantial changes to the manuscript and added a full general discussion. The paper is much improved, and I have only a few remarks.
Line 72: perhaps use a different word than artificial – or just delete.
Line 87: includes instead of identifies.
Lin 132: delete “to humans”
Line 185-186. Delete and write. Hence, we examined 20 papers.
Lin 210-212. The information should be stated in the table legend.
Line221-222. The information should be stated in the table legend.
Line 177 : do you mean 2010 – as in the period of the review ?
Line 395: Add a reference.
Line 559: delete empty line.
I trust that the editors will ensure that the tables remain within a single printed page.
